# From Isolation to Containment: Perceived Fear of Infectivity and Protective Behavioral Changes during the COVID-19 Vaccination Campaign

**DOI:** 10.3390/ijerph18126503

**Published:** 2021-06-16

**Authors:** Arielle Kaim, Maya Siman-Tov, Eli Jaffe, Bruria Adini

**Affiliations:** 1Department of Emergency and Disaster Management, Faculty of Medicine, School of Public Health, Sackler Tel Aviv University, P.O. Box 39040, Tel Aviv 6139001, Israel; ariellekaim@mail.tau.ac.il (A.K.); maylev90@gmail.com (M.S.-T.); 2Israel National Center for Trauma & Emergency Medicine Research, The Gertner Institute for Epidemiology and Health Policy Research, Sheba Medical Center, Tel Hashomer, Ramat-Gan 5266202, Israel; 3Public Relations, Training and Volunteers Division, Magen David Adom, Igal Alon 70, Tel Aviv 6706215, Israel; Eliy@mda.org.il

**Keywords:** COVID-19, pandemic, protective behavior, practices, vaccinations, infectivity

## Abstract

In the ongoing COVID-19 pandemic, maintenance of protective behavior is a continued challenge in the effort to contain the spread of the virus. A cross-sectional study via an internet questionnaire was utilized to elucidate changes in compliance to protective behavior among the Israeli population (*n* = 1120), after the beginning of the vaccination campaign. Comparison was made between individuals who were previously infected with the virus, those who received one dose of inoculation with the vaccine, and individuals that were neither infected or vaccinated. The study results indicate that those who were previously infected with the COVID-19 virus were less careful about mask wearing (18.8%) and social distancing (29.7%), as compared to the other examined groups (regarding mask wearing, 8.2% and 11.6% respectively, and with regard to social distancing 12.8% and 19.2%), and may require targeted risk communication campaigns to address this population. Furthermore, the study revealed that those that were non-Jewish (as compared to Jewish study counterparts) or that were older (19+) were more vigilant in their protective behavior (29.6% vs. 11.2% respectively for social distancing and 29.6% vs. 11.1% respectively for mask wearing). Despite a successful initial vaccination campaign in Israel, public health officials need to engage all members of the public to unremittingly observe compliance to directed health guidelines, to ensure that the results of previous governmental efforts in fighting the pandemic (such as lockdowns) will be effectively sustained, and the road to containment will be hastened.

## 1. Introduction

The novel Coronavirus disease (COVID-19), which originated in Wuhan, China in December 2019, has rapidly made a significant impact on the global community by impacting health, economic, societal, and political systems of all countries, resulting in massive morbidity and mortality [1]. This novel and emerging illness has challenged authorities with containing the spread of the virus and has resulted in the issuing of unprecedented public health measures, with social distancing regulations put in place including full lockdowns [2]. Parallel to containment efforts, numerous laboratories and research teams strived to develop and distribute candidate COVID-19 vaccines [3]. These efforts are viewed by the World Health Organization (WHO) as an essential step for the management of this pandemic and for transitioning back from the crisis status to normalcy [4,5]. Aside from providing individual, direct immunity to those who were vaccinated, vaccines have been shown to also extend curtailing infection among individuals who have not received a vaccination. This materializes when a sufficient proportion of the population uptakes the vaccine, resulting in containment of the virus and a potential way out of the pandemic [6].

While international efforts to fast-track availability of an effective vaccine are underway, stopgap measures such as tracking and isolation, together with mask wearing and physical distancing have been employed to curb the spread of the virus [7]. The Israeli Ministry of Health has been uninterruptedly releasing and updating guidelines to relay instructions to the public on necessary protective behavior during the outbreak [8]. Affected countries have exhibited substantial variation in uptake and compliance to preventive behaviors among the general public [9]. As an example, findings from Italy by Casini and Roccetti (2020) indicate that the resurgence of the virus, resulting in the second wave, may be a result of incongruous behaviors by a portion of Italian residents in observing recommended health measures, despite the government’s prevention information campaigns [10]. Baum et al. (2021) further elucidate the relevance of shaping the variation in the uptake of preventative behavior and, accordingly, that public health outcomes depend on the crucial role of political leadership and ideology, demonstrated by the varying strategies taken by governments and the varying consequences [11].

According to Levkovich and Shinan-Altman (2020), the Israeli public demonstrated high levels of compliance to public health guidelines despite variation among age groups [10]. Consistent with additional findings, adolescents and young adults have been identified as groups with lower compliance rates to public health measures [12,13,14]. Compliance was also shown to improve in phase two of the study (23 April to 5 May 2020) as compared to phase 1 (12–21 March 2020) [12]. Beyond these variables, compliance with health directives was determined to be higher in conjunction with greater negative emotional reactions (distress, fear, and stress) [12,13].

One feature of viral infection outbreaks is the fear that they can instill across large portions of the population [15,16]. Globally, heightened levels of psychological distress have been reported as a result of the COVID-19 pandemic [17,18,19]. It has previously been demonstrated that motivation to engage in protective behaviors and comply with health guidelines is influenced by perceived risk and levels of distress [18,19,20]. Israeli citizens, like their counterparts around the world, have been significantly impacted by the major public health, economic, social, and safety implications of the pandemic resulting in expressed concern, fear, and stress regarding COVID-19 [12,21,22,23]. A longitudinal study by Kimhi et al. (2020) presented an increase between the end of the pandemic’s “first wave” and the beginning of the “second wave” in distress symptoms (sense of danger, anxiety, depression, and perceived threats), and a decrease in individual, community and national resilience, among the Jewish Israeli population [23].

Flattening the emotional distress curve of COVID-19 partially may be achieved by a viable vaccine [24]. In the span of a year, Israel has gone from multiple lockdowns to being a global leader in the efficiency of its COVID-19 inoculation rollout [25]. To confer adequate immunity, at least two doses of the Pfizer or Moderna COVID-19 vaccines are necessary [26]. Israel’s citizens are at the forefront of the world’s most aggressive vaccination campaign, which began on 20 December 2020, with more than 40% of its population having received the first dose of the vaccine and 14% having had the second as of 26 January 2021 [27]. Israel’s position as a nation with the most per capita doses of the vaccine administered has become a source of national pride, potentially further encouraging the public to uptake the vaccine [28]. In spite of the initial successful national vaccination campaign, as of February 2021 the pandemic in Israel remained severe, with high concern for the emergence of new, highly contagious variants [27].

Resulting from the vaccination campaign, a false sense of protection or security may arise among the general public [29]. It has been found that infectivity may occur both in the process of vaccination as well as following it [30]. Accordingly, continued vigilance in the personal protective behavior (i.e., mask-wearing and physical distancing) of the public is essential, particularly after the first dosage of the vaccine [31]. As these vaccines are newly licensed, allowing time for vetting of safety and efficacy is of great importance in order to move towards the resumption of life pre-pandemic [32]. Early findings in Israel from the vaccination program suggest that the rollout of the Pfizer BioNTech vaccine is leading to fewer new infections and is at least 50% effective 13 to 24 days after the first dose [33]. An additional group of researchers reanalyzed the same initial data set, and claimed that the efficacy after the first dosage is close to “zero at day 14 but then rose to about 90% at day 21 before levelling off” [34]. The researchers suggest that the initial surge in infection risk is still not fully understood but may potentially be related to less cautious protective behaviors after uptake of the initial dosage [34].

The aim of the study is to investigate levels of perceived concern from COVID-19 infectivity as well as protective behavioral adjustments (i.e., mask wearing and social distancing) during the initial phase of the vaccination campaign using an internet questionnaire. To our knowledge, this is the first study to evaluate changes in protective behavior or attitudes during the vaccination program.

## 2. Materials and Methods

### 2.1. Study Procedure

This cross-sectional study was conducted in Israel during January 2021, in the midst of both an “intense lockdown” and an active vaccination campaign. A sample from the Magen David Adom Corona Ambassador Program was engaged in this study to assess self-reported attitudes and protective behavioral changes as a result of the COVID-19 vaccination campaign. The study made use of the Google Forms software, which provides a free, web-based platform for administrating surveys (https://www.google.com/forms/about/, accessed on 25 April 2021).

### 2.2. Target Population

The survey target population for this study were amongst the 2000 volunteer participants in the Magen David Adom (Israeli Emergency Medical Services) Coronavirus Ambassador program. This program provides a general introduction to the coronavirus and teaches participants how to break the chain of infection, basic hygiene to mitigate infection, and how to cope with effects of social restrictions. The free course was developed by Magen David Adom in coordination with the Education Ministry, the Workers Union, Army Radio and other agencies. The course was offered in Hebrew and Arabic.

The sample size was determined based on OpenEpi (https://www.openepi.com/SampleSize, accessed 25 April 2021), requiring 385 respondents. The study was conducted using a sample of 1120 participants who all consented to participate in the research. To partake in the study, the participants had to confirm their willingness to voluntarily participate in the research. The data was collected anonymously, following approval of the Research Committee of Magen David Adom, from 1 January 2021.

### 2.3. The Study Tool

The survey contained a brief introduction, which provided information on the background, objective, procedure, voluntary nature of participation, and declarations of anonymity and confidentiality.

The questionnaire consisted of six parts, based on items that were developed specifically for this study. The components of the questionnaire consisted of the following: (1) apprehension towards COVID-19 vaccination ranked by a 5 point Likert scale, scaling from 1 = no apprehension at all, 5 = very apprehensive; (2) history of illness with COVID-19 (Yes/No); (3) history of vaccination with COVID-19 (Yes/No); (4) change in perceived fear towards COVID-19 after the beginning of the vaccination campaign ranked by a 5 point Likert scale, scaling from 1 = no longer afraid to 5 = fear much more; (5) change in protective behavioral practices (social distancing and mask wearing) ranked by two separate items measured on Likert scales, scaling from 1 = no longer careful, 5 = much more careful; (6) demographics, assessed by 5 items including age category, gender, marital status, religion, and degree of religiosity.

### 2.4. Statistical Analysis

Descriptive statistics were used for describing the participants’ demographic characteristics (frequency, mean, and standard deviation). We recoded variable change in perceived fear towards COVID-19 and protective behavioral practices (social distancing and mask wearing) after the beginning of the vaccination campaign, into three categories: Less (1–2), Same (3) and More (4–5). Chi-square tests were used to evaluate differences between groups. A logistic regression was used to identify factors that predicted the dependent variables of fear of infectivity, maintaining social distancing and mask wearing. The variables entered to the model were gender, age, ethnicity, vaccination, fear of infectivity and ‘neither vaccinated nor infected’. All statistical analyses were performed using SPSS software version 25. *p*-values lower than 0.05 were considered to be statistically significant.

## 3. Results

### 3.1. Demographic Characteristics

The total study population included 1120 individuals that volunteered to take part in the Coronavirus Ambassador program. Half of the participants were aged 19–55, 50% were male, about 37% were married, while 58% were single. Eighty-four percent were Jews and half of the participants defined themselves as secular.

### 3.2. Vaccinated and COVID-19 Characteristics

Among those who were vaccinated (*n* = 460), seven were later infected (*n* = 7), thus they were excluded from the analysis. The study population was split into three categories: infected (*n* = 64), vaccinated (*n* = 453), neither infected nor vaccinated (*n* = 603).

Among the total study population (*n* = 1120), 517 (46.2%) either had COVID-19 or were vaccinated compared to 603 (53.8%) who were not infected with COVID-19, nor did they receive the vaccine.

Seventy-three percent of those vaccinated were Jews, compared to 27% non-Jews. In addition, among Jews, 5% were infected compared to 9% non-Jews (χ^2^ = 17.95 *p* < 0.001). From the varied age groups, among individuals aged 56 and above, 80% were vaccinated compared to 59% among the younger age group of 19–55 and 10% among the age group that is <=18. In addition, 4% among the age group of 56 and above were infected compared to 6% among the age group of 19–55 and 6% among the age group that is 18 and below (χ^2^ = 342.5 *p* < 0.001). The full characteristics of the study population are presented in Table 1.

### 3.3. Protective Behavioral Practices and Perceived Fear towards COVID-19

Among the total sample, after the beginning of the vaccination campaign, about a third (35%) reported less perceived fear towards COVID-19, 51% had no change in fear and 15% were more afraid. Regarding social distancing, 14% were more watchful towards social distancing, 17% were less careful and 69% reported no change. Regarding personal protection, 14% were more careful with mask wearing, 10% were less and 75% with no change.

### 3.4. Differences between Protective Behavior to Being Vaccinated, Infected or Neither of Them

Figure 1 and Figure 2 present variability in behavioral practices (social distancing and mask wearing) according to the individuals’ affiliation to one of the three groups: vaccinated, infected and those that fell into neither vaccinated nor infected category. It appears that following the beginning of the vaccination campaign, those who were infected were less careful about social distancing (29.7%) compared to those who received the first dose of vaccine (12.8%), or those who were neither vaccinated nor infected (19.2%) (χ^2^ = 19.32 *p* = 0.001). See Figure 1.

Consistently, after the beginning of the vaccination campaign, those who were infected were less careful about mask wearing (18.8%) as compared to those who only received the first dose of the vaccine (8.2%) and those who neither were vaccinated nor infected (11.6%) (χ^2^ = 13.02 *p* = 0.011). See Figure 2. No association was found between levels of perceived fear of COVID-19 and belonging to one of the three groups—vaccinated, infected or neither of them (χ^2^ = 3.50 *p* = 0.478).

### 3.5. Fear of Infectivity Following the Beginning of the Vaccination Campaign According to Ethnic Affiliation

We found a higher percentage of non-Jews reporting higher levels of fear from COVID-19 infection after the beginning of the vaccination campaign as compared to Jews (25.7% vs. 12.5% respectively; χ^2^ = 27.45 *p* < 0.001). The higher levels of fear among non-Jews was maintained even when we analyzed the data according to the three groups: those who were vaccinated (25.0% among non-Jews vs. 12.1% among Jews; χ^2^ = 9.13 *p* = 0.010); those who were infected (21.3% among non-Jews and 10.4% among Jews; χ^2^ = 6.24 *p* = 0.040); and those who were neither vaccinated nor infected (21.3% among non-Jews and 10.4% among Jews; χ^2^ = 6.24 *p* = 0.040). (Figure 3).

### 3.6. Change in Protective Behavior Following the Vaccination Campaign according to Ethnic Affiliation

A higher percent of non-Jews reported paying more attention to social distancing after the beginning of the vaccination campaign, as compared to Jews (29.6% vs. 11.2% respectively; χ^2^ = 50.06 *p* < 0.001). The higher level of social distancing among Non-Jews was maintained even when we analyzed the data according to the three groups: those who were vaccinated (41.7% among non-Jews vs. 14.3% among Jews; χ^2^ = 22.9 *p* < 0.001), those who were infected (33.1% among non-Jews vs. 8.3% among Jews; χ^2^= 7.29 *p* = 0.026) and those who were neither vaccinated nor infected (24.3% among non-Jews and 8.8% among Jews; χ^2^ = 28.81 *p* < 0.001). See Figure 4.

A higher percent of non-Jews reported paying more attention to mask wearing after the beginning of the vaccination campaign as compared to Jews (29.6% vs. 11.1% respectively; χ^2^ = 52.90 *p* < 0.001). The higher level of mask wearing among non-Jews was maintained even when we analyzed the data according to the three groups: those who were vaccinated (33.3% among non-Jews vs. 14.6% among Jews; χ^2^ = 12.04 *p* = 0.002), those who were infected (37.5% among non-Jews vs. 10.4% among Jews; χ^2^= 7.91 *p* = 0.019) and those who were neither vaccinated nor infected (27.0% among non-Jews and 8.2% among Jews; χ^2^ = 36.67 *p* < 0.001). See Figure 5.

### 3.7. Change in Protective Behavior Following the Vaccination Campaign According to Age

Concerning age groups, we found a higher percentage of the age groups 19–55 and 56 and above reporting that they paid more attention to mask wearing after the beginning of the vaccination campaign, as compared to the younger age group of 18 or less (16.5%, 17.6%, and 10.4% respectively; χ^2^ = 11.28 *p* = 0.024). A higher percentage of the age group 19–55 and 56 and above reported paying more attention to social distancing after the beginning of the vaccination campaign, as compared to the age group of less than 18 (17.9%, 17.6%, and 8.9% respectively; χ^2^ = 23.84 *p* < 0.001). No association was found between age group and fear from COVID-19 infection after the beginning of the vaccination campaign.

### 3.8. Prediction of Fear of Infectivity and Protective Behavior

The logistic regression did not identify any factor that significantly predicted fear of infectivity. Two factors of ethnicity and age predicted 9.4% of social distancing; Arab versus Jewish Israelis had a 3.72-fold probability of maintaining social distancing, while young (≤18) versus older individuals had a 0.4 (lower) probability of maintaining social distancing. Similar findings were found concerning mask wearing; ethnicity and age predicted 8.3% of mask wearing. Arab versus Jewish Israelis had a 3.73-fold probability of wearing masks, while young (≤18) versus older individuals, had a 0.5 (lower) probability of wearing masks. None of the regression models presented that the vaccination, the fear of infectivity, nor experiencing neither of these, better predicted a higher social distancing or mask wearing.

## 4. Discussion

Understanding the public’s perception of risk and adoption of protective behavior is essential for the purpose of mitigating the consequences of an emergency, particularly in the case of pandemics such as the COVID-19 outbreak [34,35]. As countries worldwide are continuing to face remarkable challenges in curbing the spread of the COVID-19 virus, continued focus on changes in levels of concern towards COVID-19 as well as compliance with protective behavior (i.e., mask wearing and social distancing) is essential, throughout the different phases of the pandemic [36]. Reports of rising infections with the COVID-19 virus, as well as the potential that vaccinated individuals serve as asymptomatic carriers, suggest that these individuals may still constitute a significant reservoir of future spread, putting vulnerable populations at risk with a false sense of security [37,38,39]. To ensure the effectiveness of all measures implemented until now and to achieve the aspired goal of containment, it is essential that populations observe continued vigilance in accordance with directed health guidelines [31,40]. Therefore, with this agenda in mind, the current study investigated changes in levels of concern and protective behavior (i.e., mask wearing and social distancing) amongst individuals who were previously infected, received the initial dosage of the vaccine, or neither.

The findings of this investigation demonstrate several interesting phenomena. First, the results of the study suggest that those who were infected became less vigilant about protective behavior, both concerning social distancing and mask wearing, after the beginning of the vaccination campaign. Evidence from previous infectious disease outbreaks, and the ongoing pandemic, points to the role of higher levels of perceived personal and familial threat of infection, as well as higher perceived severity of the outbreak for engagement in protective behaviors, including in the intention or uptake of vaccinations [12,13,41,42,43]. Individuals who have previously been infected and successfully recovered from the virus may have a diminished sense of perceived risk and perceived severity of the virus as exposure has already been confronted. It is widely discussed in the literature that novel risks often induce fear; however, after repeated exposure, arousal of fear is diminished, thus resulting in underestimation of the risk, prompting laxer risk behaviors [42,43]. In addition, throughout the early stages of the pandemic, key questions concerning the course of the COVID-19 pandemic involved how well and for how long the immune responses protect the individual from reinfection [44]. Only after growing evidence from surfacing case reports did the scientific community come to understand this phenomenon as an additional hurdle to achieving containment [45]. The delayed recognition of potential reinfection as a phenomenon may have congruently contributed to the lower levels of perceived risk among this population.

Furthermore, the results of this study indicate that those that went to get vaccinated reported stricter adherence to protective behavior. As described above, an individual acceptance of vaccination suggests a higher general awareness towards the risk of the virus, and consistently this awareness can offer potential pathways for higher levels of engagement in additional protective behavior [46,47]. This is reflected in our results through the increased stringency of mask-wearing and social distancing amongst this population. The behavioral adaptations for avoiding infection similarly may be indicative of the fact that this group, at the time of the investigation, were in-between receiving their first and second doses in order to achieve a state of full inoculation. Given that it was widely conveyed to the public that the initial dosage of the vaccine may not provide an adequate level of immunity as well as suboptimal long-term disease protection, this awareness may have prompted additional precautionary behavior by this study group [30,33,34].

The findings concerning the non-Jewish population and their higher levels of perceived fear towards infection with COVID-19 and the corresponding increased reported protective behavior as compared to the Jewish population, are in line with previous findings by Braun-Lewensohn (2020) and Kimhi et al. (2020), which indicated that highest levels of emotional distress and fear as a result of the virus were found among the non-Jewish population in Israel during the COVID-19 pandemic [47,48]. These findings were further reflected in the reported levels of uptake to social distancing and mask wearing and are in line with the findings of Bodas and Peleg (2020), which revealed that non-Jewish respondents who were more worried about COVID-19 were more likely to comply with self-isolation measures than their respective counterparts [49]. The data available concerning mortality and morbidity rates in Israel reflect that the Arab sector was more highly afflicted as compared to the general Jewish population, a possible explanation for their increased fear concerning infection and increased protective behavior [50]. In addition, differences in behavior between these populations may be a function of the unique backdrop of additional contributing factors, such as disparities in cultural, social, economic, and national status which must be considered [48]. Lastly, in line with our findings concerning age, older age has been widely demonstrated to be a predictive factor for more dutiful compliance to protective behavior [51,52,53]. Alongside the fact that older age was one of the most important factors in diminishing one’s chances of surviving COVID-19, older populations were found to more likely perceive COVID-19 as a significant crisis, and thus exhibited more self-preservatory behavior [52]. Nonetheless, as there is some evidence that robust spread of COVID-19 occurred in secondary/high schools, younger populations must also actively participate in social distancing and mask wearing efforts [54]. Considering the findings of this study, according to which the younger population (that was not at the time of the study vaccinated) displayed lower levels of concern from being infected, as well as lower levels of compliance with protective behavior, strict attention should be directed towards this group.

Several main limitations have been identified with regard to this study. The first is that participants were asked to self-report compliance levels to protective behavior, and thus the actual changes in compliance cannot be verified. As in all studies based on questionnaires, social desirability bias cannot be ruled out with regard to the results identified. In addition, because this study was conducted cross-sectionally, via the internet, the study conclusions are limited to the given time point at which the information was collected and to persons who have access to a source of internet and high computing skills. The respondents represent a convenience sample. The sample population included only 5.7% of individuals infected by COVID-19; though this is a relatively small sample, the analysis of the study presented significant results concerning the different reported protective behavior of this population compared to individuals that were either vaccinated or ‘not infected and not vaccinated’. Lastly, because this study was conducted in Hebrew, members of the Israeli population who are not fluent in the language were unable to participate in this study.

## 5. Conclusions

To conclude, the obtained observations from this questionnaire study are of utmost importance as they improve our understanding and response to the COVID-19 pandemic with clear implications and normative concerns over the reliability of protective behavior among those that recovered, received vaccination, or those who do not fall into either category. In particular, to address the over-confidence exhibited by those who were previously infected, targeted health communication strategies may be necessary to address misconceptions about possible reinfection and the possibility of serving as asymptomatic carriers [35,36,37]. In addition, to ensure continued general population compliance to health directives and continued curtailing of the pandemic, guidelines and information campaigns must reemphasize that the waged war against COVID-19 is not yet over. As Israel reopens the country after its third lockdown and eases restrictions, this continued compliance to protective behavior among the public will be critical to Israel’s sustained success in the fight against the pandemic and in the road to containment.

## Figures and Tables

**Figure 1 ijerph-18-06503-f001:**
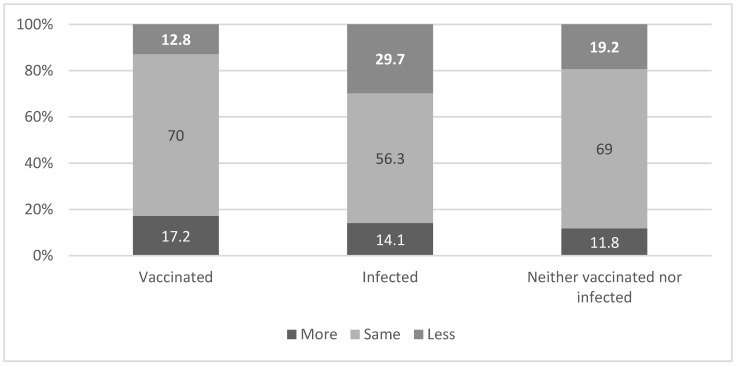
Level of change in social distancing after the beginning of the vaccination campaign by vaccinated and infected variable. χ^2^ = 19.32 *p* = 0.001.

**Figure 2 ijerph-18-06503-f002:**
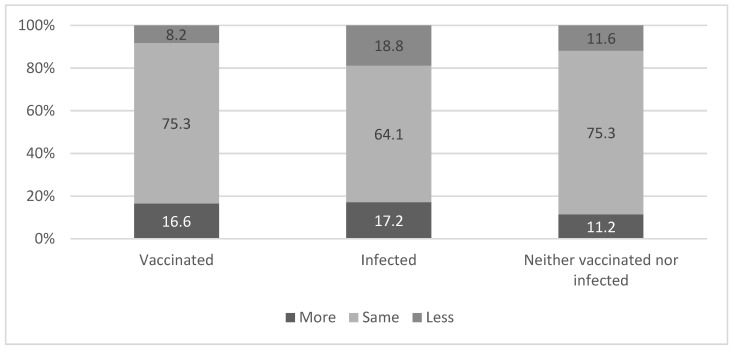
Level of change in mask wearing after the beginning of the vaccination campaign by vaccinated and infected variable. χ^2^ = 13.02 *p* = 0.011.

**Figure 3 ijerph-18-06503-f003:**
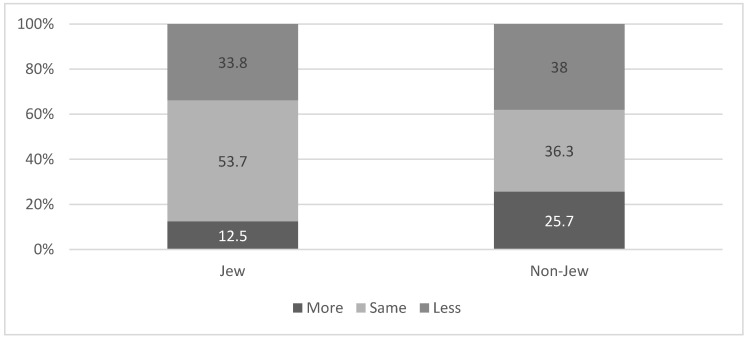
Fear of being infected with the Coronavirus following the beginning of the vaccination campaign, according to ethnic affiliation. χ^2^ = 27.45 *p* < 0.001.

**Figure 4 ijerph-18-06503-f004:**
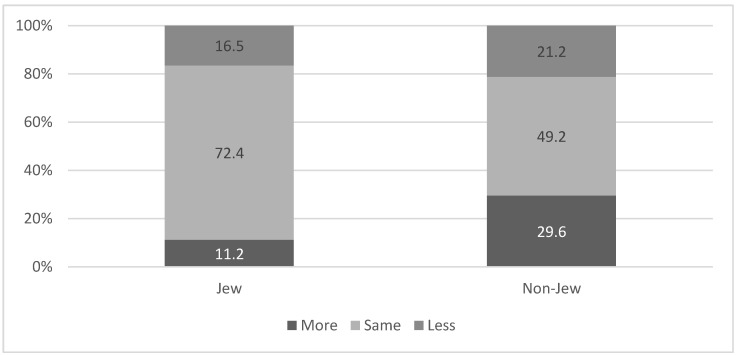
Change in social distancing following the vaccination campaign, according to ethnic affiliation. χ^2^ = 50.06 *p* < 0.001.

**Figure 5 ijerph-18-06503-f005:**
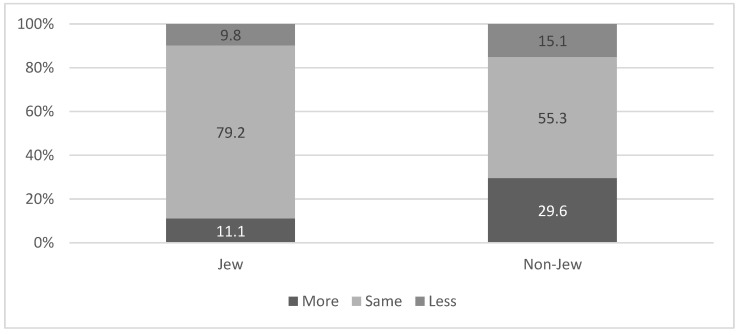
Change in mask-wearing following the vaccination campaign, according to ethnic affiliation. χ^2^ = 52.90 *p* < 0.001.

**Table 1 ijerph-18-06503-t001:** Study population, n = 1120.

	N	%
Age		
<=18	471	42.1%
19–55	547	48.8%
56+	102	9.1%
Sex		
Male	566	50.4%
Female	554	49.6%
Marital status		
Married	422	37.7%
Single	648	57.9%
Divorced/widowed	49	4.4%
Religion		
Jew	941	84.0%
Non-Jew	179	16.0%
Level of religiosity		
Secular	562	50.2%
Religious	558	49.8%
COVID-19 infected		
Yes	64	5.7%
No	1056	94.3%
COVID-19 vaccine		
Yes	460	41.1%
No	660	58.9%

## Data Availability

The dataset used for this study is available to the authors of this article. It is not stored publicly due to ethical and privacy issues.

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
