# Peer review of "From Isolation to Containment: Perceived Fear of Infectivity and Protective Behavioral Changes during the COVID-19 Vaccination Campaign"

_ijerph, 2021, doi:10.3390/ijerph18126503_

Round 1

Reviewer 1 Report

The MS “From Isolation to Herd Immunity: Perceived Fear of Infectivity and Protective Behavioral Changes During the COVID-19 Vaccination Campaign” submitted by Kaim et al. is an important contribution to understand varying perceptions and behaviors in the population during the ongoing COVID-19 pandemic.

  1. Title and Abstract (line 25): I do propose to refrain from speaking of herd immunity. For several reasons herd immunity cannot be reached in the sense this is the case e.g. for measles. 1) neither infection nor vaccination results in full protection from (re-)infection; 2) vaccination or infection does not prohibit transmission; 3) this is aggravated by newly emerging variants. If a certain level of vaccination is reached, it will be easier to contain the now and then occurring clusters of infection. But this is all that can be hoped for in the near future.

  1. Introduction (lines 42/43): Same as above. The paper cited (Ref 6) is a commentary that was written long before any vaccine was available and at a time we did not know whether the vaccines will provide sterilizing immunity. Now we know that this is not the case and therefore we cannot reach herd immunity.

  1. Abstract: Important information is missing: Number of participants and that data were obtained by an internet questionnaire must be mentioned. Furthermore, present some important results quantitatively.

  1. Discussion (lines 303-305): The anti-body levels do not play a major role in protection from COVID-19. The two shots are not necessary for giving full protection as it is sometimes conveyed to the public, but to counteract waning immunity and to provide longer protection. In fact, the second dose does not change the level of protection in any way as is the case for inactivated virus vaccines that have a basic immunization course consisting of several doses. The mRNA vaccines provide protection from about day 22 onwards whether given a second dose or not. But immunity would decline during the next about three to five months to levels that provide no longer high levels of protection. This waning immunity is less pronounced by giving the second dose. More precise data about the duration of protection will be available in the near future.

  1. Discussion (line 307/308): You have not assessed behavior but reports about behavior!

  1. Discussion: According to published time trends of SARS-CoV2 infection, COVID-19 hospitalizations and deaths in Israel, during the first months of the vaccination campaign, levels were rising first and declining only after some time. It has been suggested that this time-trend was due to vaccinees caring less about protection measures. The results of this study indicate that there is no difference in this regard between participants that were already vaccinated and those neither vaccinated nor infected. Authors should discuss this discrepancy. Maybe a multivariate analysis using ordinal regression and including age, sex and ethnicity among predictors and not only vaccination/infection status would reveal determinants of protective behavior that could better explain the findings.  

Minor errors:

Line 33: Omit ‘rates’ (rates are low but numbers are high!)

Line 76: Place a comma after campaign

Line 85: Delete ‘e’ in Accordingly

Line 120: Insert the number instead of ----

Line 125ff: Section 2.3 is duplicated

Line 176: The figure ‘43%’ must be 73%

Lines 179, 181: Replace >18 by <=18 or 18 and below

Table 1: Replace <18 by <=18

Line 286: Omit comma after laxer

Reviewer 2 Report

This paper has a potential for publication, nonetheless some issues should be better addressed before I can express a full support.

Being the main topics of the paper quite clear (self-protective behavior of a population against Covid-19, mostly with face-masking and social distancing), the first issue is that some of the behaviors observed by authors should not be confined to Israel, only.

While I can accept that this study investigates Israel only, for various technical motivations, I urge the authors to extend their Introduction to include a more detailed discussion concerning other geographies.

For example, these two following facts are well known in relation to the attitude towards face masking and social distancing:

1 Italy summer 2020, the spread of the disease seemed to be under control and most Italians went on vacation, without any or little respect towards containment measures and this failure in the social behavior was one of the main motivation recognized behind the start of the second wave, as discussed at length here (to be discussed and cited):

- A Cross-Regional Analysis of the COVID-19 Spread during the 2020 Italian Vacation Period: Results from Three Computational Models Are Compared, Luca Casini and Marco Roccetti, Sensors 2020, 20(24), 7319   2 Political leadership. Of relevance has been tthe role that political leadership and ideology plays in shaping public health responses. For example New Zealand Prime Minister exerted a strong political leadership with her empathic and clear communication to the public yielding positive results. On the contrary, both US (with the former administration) and Brazil, failed to accept scientific public health advice, including mask wearing and social distancing, yielding very negative results. All this is discussed in the following paper (to be discussed and cited):   - Explaining covid-19 performance: what factors might predict national responses? Fran Baum et al. BMJ 2021; 372: n91   A second point is about the scientific process the authors of this research has followed. Using questionnaires can be a plausible tool, yet the author should never confound this toll with a natural observational experiment; hence I urge them to emphasize more and better (even in the Introduction and in the Conclusions) that the obtained results have been derived with such a methodology.   A third fact that put me in difficulty is the limited number of infected people investigated that are really few with respect to the vaccinated (453) and sane (>600). This is a crucial point, posing a potential statistical problem. My opinion is that the number of infected people should be enlarged for the study to be statistically significant.   A final consideration pertains the portion of the study that investigates on Jews and non-Jews. Maybe there are several, complex motivations at the basis of different behavior (politics, low income, ...) that should be better investigated or at least discussed. I understand this is a delicate issue, nonetheless a scientific study needs such an investigation. 

Round 2

Reviewer 2 Report

Paper has greatly improved. It can be pubhlished